# Ezrin Contributes to the Plasma Membrane Expression of PD–L1 in A2780 Cells

**DOI:** 10.3390/jcm11092457

**Published:** 2022-04-27

**Authors:** Mayuka Tameishi, Honami Ishikawa, Chihiro Tanaka, Takuro Kobori, Yoko Urashima, Takuya Ito, Tokio Obata

**Affiliations:** 1Laboratory of Clinical Pharmaceutics, Faculty of Pharmacy, Osaka Ohtani University, Tondabayashi 584-8540, Japan; u4117083@osaka-ohtani.ac.jp (M.T.); u4118007@osaka-ohtani.ac.jp (H.I.); u4117078@osaka-ohtani.ac.jp (C.T.); koboritaku@osaka-ohtani.ac.jp (T.K.); urasiyo@osaka-ohtani.ac.jp (Y.U.); 2Laboratory of Natural Medicines, Faculty of Pharmacy, Osaka Ohtani University, Tondabayashi 584-8540, Japan; itoutaku@osaka-ohtani.ac.jp

**Keywords:** programmed death ligand–1, immune checkpoint molecules, ezrin, radixin, moesin, ovarian cancer, cancer immunotherapy

## Abstract

Programmed death ligand–1 (PD–L1) is one of the immune checkpoint molecule localized on the plasma membrane of numerous cancer cells that negatively regulates T-cell-mediated immunosurveillance. Despite the remarkable efficacy and safety profile of immune checkpoint inhibitors (ICIs), such as anti-PD–L1 antibodies, restricted poor therapeutic responses to ICIs are often observed in patients with ovarian cancer. Because higher expression of PD–L1 in advanced ovarian cancer is associated with a decreased survival rate, identifying the potential molecules to regulate the plasma membrane expression of PD–L1 may provide a novel therapeutic strategy to improve the efficacy of ICIs against ovarian cancers. Here, we reveal the involvement of the ezrin/radixin/moesin (ERM) family, which crosslinks transmembrane proteins with the actin cytoskeleton by serving as a scaffold protein, in the plasma membrane expression of PD–L1 in the human epithelial ovarian cancer cell line A2780. Our results demonstrate that PD–L1 and all three ERMs were expressed at the mRNA and protein levels in A2780 cells, and that PD–L1 was highly colocalized with ezrin and moesin, but moderately with radixin, in the plasma membrane. Interestingly, RNA interference-mediated gene silencing of ezrin, but not of radixin or moesin, substantially reduced the plasma membrane expression of PD–L1 without altering its mRNA expression. In conclusion, our results indicate that ezrin may be responsible for the plasma membrane expression of PD–L1, possibly by serving as a scaffold protein in A2780 cells. Ezrin is a potential therapeutic target for improving the efficacy of ICIs against ovarian cancers.

## 1. Introduction

Ovarian cancer is considered the most lethal gynecological cancer and the eighth leading cause of cancer-related death and incidence in females worldwide [1] with a high heterogeneity in morphology, molecular alterations, and clinical behavior [2]. Ovarian cancer is morphologically categorized into two major types: non-epithelial and epithelial groups, the latter of which accounts for the vast majority of ovarian cancer cases [3]. Based on the clinicopathologic factors, epithelial malignancy is further divided into two biologically distinct groups: high-grade serous ovaria carcinoma (HGSOC), responsible for more than 70% of ovarian cancer death and over 80% of advanced-stage ovarian cancer diagnoses, and non-HGSOC [2]. Because of the lack of symptoms and screening techniques for diagnosis in the early stages, ovarian cancer is often diagnosed at an advanced stage, with a 5-year survival rate of only 30% [4]. Furthermore, most patients experience recurrence within a few years after the primary surgery, followed by platinum- and taxane-based chemotherapy, leading to an extremely low survival rate [3]. Therefore, it is important to develop effective alternative treatment strategies.

Programmed death–1 (PD–1) and programmed death ligand–1 (PD–L1) are immune checkpoint proteins that negatively regulate the T-cell immune system. PD–L1 is expressed on the plasma membrane of numerous cancer cells and macrophages in tumor tissues [5,6], and interacts with PD–1 expressed on activated cytotoxic T-cell membranes, which leads to an apoptosis of activated T cells, thereby blocking antitumor T-cell immunity [7,8,9]. Intriguingly, higher PD–L1 expression in advanced ovarian cancer cells is associated with a decreased survival rate [10]. Recently, immune checkpoint inhibitors (ICIs) targeting PD–L1 and PD–1 have led to significant survival benefits in various advanced cancers by reactivating antitumor immunity displayed by T cells [11,12]. Despite the remarkable therapeutic efficacy and acceptable safety profile of anti-PD–1/PD–L1 antibodies (Abs) in a small subset of patients with platinum-resistant, recurrent, or refractory advanced ovarian cancer, restricted therapeutic responses to ICIs are often observed in patients with ovarian cancer [13,14]. In addition, only 10% of patients respond clinically to anti-PD–1/PD–L1 Abs, owing to early or acquired resistance [15]. Although tumor genomic, transcriptomic, and immune profiling were characterized in patients with HGSOC [16,17], it has yet to be determined whether certain subtypes of ovarian cancer are more sensitive to ICIs. Therefore, novel therapeutic targets need to be identified.

Accumulating evidence indicated that post-translational modifications, including phosphorylation, glycosylation, and ubiquitination, strongly influence the plasma membrane localization of PD–L1 as a transmembrane protein [18,19,20,21,22]. In particular, our laboratory has focused on the family of ezrin/radixin/moesin (ERM) proteins, which has a high degree of sequence similarity with each other that crosslink the actin cytoskeleton to several cancer-related transmembrane proteins, including transmembrane receptor kinases and drug transporters through direct molecular interaction [23,24,25,26]. Interestingly, we recently reported that the ERM family post-translationally regulates the cell surface localization of PD–L1 by serving as scaffold proteins in some types of human cancer cell lines [27,28,29,30,31]. However, whether the ERM family modulates the plasma membrane expression of PD–L1 as a scaffold protein in ovarian cancer cells has yet to be determined.

The aim of this study was to examine the role of ERM proteins in the plasma membrane expression of PD–L1 in A2780 cells, derived from human epithelial ovarian cancer.

## 2. Materials and Methods

### 2.1. Cell Culture

The human epithelial ovarian cancer cell line A2780 was obtained from the European Collection of Authenticated Cell Cultures (ECACC) (EC93112519–F0; KAC, Hyogo, Japan). A2780 cells were grown in Roswell Park Memorial Institute (RPMI)–1640 medium (FUJIFILM Wako Pure Chemical, Osaka, Japan) supplemented with heat-inactivated 10% fetal bovine serum (BioWest, Nuailé, France) at a passage number from 8 up to 18. Cultures were maintained at 37 °C in a humidified atmosphere containing 5% CO_2_.

### 2.2. siRNA Treatment

A2780 cells seeded in 24-well plates at a density of 2.0 × 10⁴ cells/well were transfected with silencer select small interfering (si) RNAs for human ezrin (siRNA ID s14795), radixin (siRNA ID s11899), moesin (siRNA ID s8986), PD–L1 (siRNA ID s26549), or those for non–targeting control (NC), at the dose of 5 nM delivered through lipofection using Lipofectamine RNAiMAX at the volume of 0.10 µL/2.0 × 10⁴ cells, and cultured continuously for 4 days without replenishing the culture medium. All reagents used for siRNA treatment were obtained from Thermo Fisher Scientific (Tokyo, Japan).

### 2.3. RNA Extraction and Real–Time Reverse Transcription–Polymerase Chain Reaction

Total RNA from A2780 cells was extracted, followed by assessment for the concentration and purity of total RNA, as described previously [27,28,29,30]. Subsequent real-time reverse transcription (RT)–polymerase chain reaction (PCR) and data analysis were performed as described previously [27,28,29,30]. The gene-specific primer sequences are provided in Appendix A.

### 2.4. Confocal Laser Scanning Microscopy Analysis

Confocal laser scanning microscopy analysis was conducted, as described previously [29,30]. After fixation, plasma membrane permeabilization, and blocking, cells were reacted with respective primary Abs. For the detection of ERM, the cells were subsequently incubated with Alexa Fluor 488-conjugated secondary Ab. Cells were then incubated with a blocking buffer containing F-actin probe (phalloidin) conjugated to tetramethylrhodamine isothiocyanate for cell membrane counterstaining, followed by confocal immunofluorescence using a Nikon A1 confocal laser microscope system (Nikon Instruments, Tokyo, Japan). For immunofluorescence double staining of ERM and PD–L1, the procedure described above was followed until the primary Ab reaction against ERM. Next, the cells were incubated with Alexa Fluor 594-conjugated secondary Ab, followed by incubation with anti-PD–L1 Ab labeled with Alexa Fluor 488 in the blocking buffer. Thereafter, photomicrographs were taken as described above. All the Abs used in the present study are provided in Appendix A.

### 2.5. Protein Isolation and Western Blotting

A2780 cells were lysed in a radioimmunoprecipitation assay (RIPA) buffer containing a protease inhibitor cocktail and heated at 97 °C with 2× Laemmli buffer. After that, proteins were separated using sodium dodecyl sulfate–polyacrylamide gel electrophoresis and transferred onto nitrocellulose membranes (Bio-Rad Laboratories, Tokyo, Japan). The membranes were blocked in 5% nonfat dry milk diluted with Dulbecco’s phosphate-buffered saline (D–PBS) containing 0.1% Tween–20 (PBS–T) for 1 h at room temperature, and subsequently probed with respective primary Abs in PBS–T containing 5% nonfat dry milk at 4 °C overnight. Blots were incubated for 1 h at room temperature. Protein bands were visualized using horseradish peroxidase (HRP)-conjugated secondary Abs and enhanced chemiluminescence (ECL) Prime Western Blotting Detection Reagent (Cytiva, Tokyo, Japan), and analyzed with a LuminoGraph II EM (ATTO, Tokyo, Japan). Whole blots to confirm the reactivity of all Abs used in western blotting are shown in Appendix A. Original source images for all data obtained by immunoblots are provided in Appendix A.

### 2.6. Flow Cytometry

Cell suspensions were treated with anti-PD–L1 Ab conjugated with an allophycocyanin (APC) in a labelling buffer consisting of 5% normal horse serum and 1% sodium azide in D–PBS for 1 h at 4 °C in the dark. Cells were subsequently analyzed using a Cell Analyzer EC800 (Sony Imaging Products and Solutions, Tokyo, Japan).

### 2.7. Statistical Analysis

Values are expressed as mean ± standard error of the mean (SEM). Statistical significance was assessed using one-way ANOVA followed by Dunnett’s test for multiple comparisons using Prism version 3 (GraphPad Software, La Jolla, CA, USA). Statistical significance was set at *p* < 0.05.

## 3. Results

### 3.1. Expression Profiles of PD–L1 and ERMs in A2780 Cells

We analyzed the gene expression patterns of PD–L1 and ERMs in human ovarian adenocarcinoma strains using the public database of the Cancer Dependency Map (DepMap) portal data explorer [32,33]. The results indicated that the mRNA expression of all three ERMs was abundant in A2780 cells and that PD–L1 mRNA expression in A2780 cells was comparable to that in other cells (Figure 1a). The protein levels of PD-L1 and ERM were also detected in A2780 cells (Figure 1b).

### 3.2. PD-L1 and ERM Were Localized in the Plasma Membrane of A2780 Cells

The subcellular distribution of PD–L1, ezrin, and moesin in A2780 cells was similar to that of F-actin, indicating the plasma membrane localization of PD–L1, ezrin, and moesin (Figure 2a,b,d). In contrast, radixin was equally distributed in both the plasma membrane and cytoplasmic regions (Figure 2c). Interestingly, PD–L1 was highly colocalized with ezrin and moesin in the plasma membrane (Figure 3a,c), whereas it was moderately colocalized with radixin in A2780 cells (Figure 3b). No fluorescence signals were detected in A2780 cells exposed to secondary antibodies conjugated with an Alexa Fluor 488, or an Alexa Fluor 594, without any primary antibodies (Appendix A).

### 3.3. Influence of ERM Gene Suppression on the Expression of PD–L1 in A2780 Cells

We checked the gene-silencing effects of siRNA against each ERM on target mRNA expression levels in A2780 cells. siRNA against each ERM significantly suppressed the expression of the target mRNA in comparison with the control (Figure 4a–c). NC siRNA treatment also significantly, but slightly, decreased the mRNA levels of all three ERM (Figure 4a–c). None of the siRNAs used in this study had any influence on the viability of A2780 cells (Appendix A). Next, we examined whether the RNA interference-mediated silencing of each ERM influenced PD–L1 mRNA expression. No alterations in PD–L1 mRNA expressions were induced by siRNAs against ezrin and radixin, whereas gene silencing of PD–L1 substantially reduced PD–L1 mRNA levels (Figure 5a). By contrast, gene silencing of moesin significantly increased PD–L1 mRNA levels (Figure 5a). Finally, we tested the influence of ERM gene suppression on surface PD–L1 expression. Flow cytometry analysis showed that ezrin knockdown significantly reduced surface PD–L1 expression to the level observed with siRNA against PD–L1 (Figure 5b,c). In contrast, suppression of radixin and moesin genes had no impact on PD–L1 expression on the plasma membrane surface (Figure 5b). Together, ezrin contributes to the plasma membrane expression of PD–L1 in A2780 cells, perhaps via a post-translational process.

## 4. Discussion

In this study, we confirmed the expression of PD–L1 and all three ERMs in A2780 cells at mRNA and protein levels, the results of which are in line with the data obtained through the DepMap portal tool. Previous studies have shown the mRNA and protein levels of PD–L1 in various human ovarian adenocarcinoma cell lines, such as A2780, SKOV3, OC316, CAOV3, and OVCAR–3 [34,35,36,37]. Similarly, a large number of evidence has demonstrated that a variety of human ovarian cancer cell lines have an abundant expression of ezrin and moesin, and that overexpression of ezrin and moesin is highly associated with reduced overall survival and the progression of metastasis in clinical ovarian carcinoma specimens [38,39,40,41,42,43,44]. In contrast, there are no reports describing the expression of radixin in human ovarian cancers, except in human ovarian granulosa cells [45]. Interestingly, we observed that PD–L1 was colocalized with ezrin and moesin in the plasma membrane of A2780 cells, while was hardly colocalized with radixin because of the intracellular distribution of radixin predominantly in the cytoplasm rather than in the plasma membrane. Taken together, PD–L1 and all three ERM proteins were detected at sufficient levels in A2780 cells, whereas PD–L1 was colocalized with ezrin and moesin, but not with radixin, in the plasma membrane.

Accumulating evidence suggests that PD–L1 expression is modulated by a variety of intracellular events. Recently, post-translational modification of PD–L1 has attracted attention, owing to its ability to regulate the cell surface localization of PD–L1 [18,19,46]. Interestingly, we recently advocated that ERM proteins interact with PD–L1, contributing to the plasma membrane expression of PD–L1 [27,28,29,30,31]. Through protein–protein interactions, ezrin modulates the plasma membrane expression of PD–L1 in human uterine cervical adenocarcinoma HeLa cells, choriocarcinoma JEG–3 cells, and colon adenocarcinoma LS180 cells, where ezrin is dominantly expressed among ERM proteins as assessed by global mRNA expression analysis using DepMap and our real-time RT–PCR results [27,28,29]. We also reported that radixin is involved in the plasma membrane expression of PD–L1 as a predominant ERM protein in KP–2 human pancreatic ductal adenocarcinoma cells [30]. In contrast, other researchers revealed the function of moesin in stabilizing the plasma membrane expression of PD–L1 by inhibiting proteasomal degradation in the human breast cancer adenocarcinoma [47]. Similarly, gene silencing of ezrin, but not radixin or moesin, considerably decreased the plasma membrane expression level of PD–L1 in A2780 cells without altering the mRNA expression level of PD–L1. To the best of our knowledge, this is the first study to demonstrate the role of ezrin in the plasma membrane expression of PD–L1 in human ovarian cancer cells. The differences in the effects of ERM proteins responsible for cell surface localization of PD–L1 might result, at least in part, from the expression profile of ERM proteins, depending on the cancer cell type. Taken together, our results imply that, among ERM proteins, ezrin primarily regulates the plasma membrane expression of PD–L1 in A2780 cells.

Surprisingly, gene silencing of moesin significantly increased the mRNA expression levels of PD–L1 in A2780 cells. One possibility is that gene silencing of moesin may facilitate the production of pro-inflammatory cytokines, including interferon (IFN)–γ, tumor necrosis factor (TNF)–α, and interleukin (IL)–6, all of which can upregulate PD–L1 mRNA expression in tumors [20,48,49]. In fact, siRNA against moesin significantly increased the mRNA expression level of IFN–γ in A2780 (Appendix A), which may, in turn, upregulate PD–L1 mRNA expression. Furthermore, other researchers have demonstrated that treatment with a neutralizing Ab against moesin drives the release of IFN–γ, TNF–α, and IL–6 from several immune cells derived from human peripheral whole blood [50,51], and that serum concentrations of IFN–γ, TNF–α, and IL–6 were higher in patients with myeloperoxidase–antineutrophil cytoplasmic Ab-associated vasculitis, who are positive for anti-moesin Ab than in those negative for anti-moesin Ab [51]. These previous studies may partly explain the present results, that siRNA against moesin increases the expression level of PD–L1 mRNA in A2780 cells.

In summary, among the three ERM proteins present in A2780 cells, ezrin may function as a scaffold protein for PD–L1, primarily responsible for the plasma membrane expression of PD–L1, presumably via the post-translational modification process. In order to ascertain whether the results of our present in vitro experiments are also replicated in in vivo tissue samples, in vivo experiments employing xenograft model mice should be performed in our future studies.

## Figures and Tables

**Figure 1 jcm-11-02457-f001:**
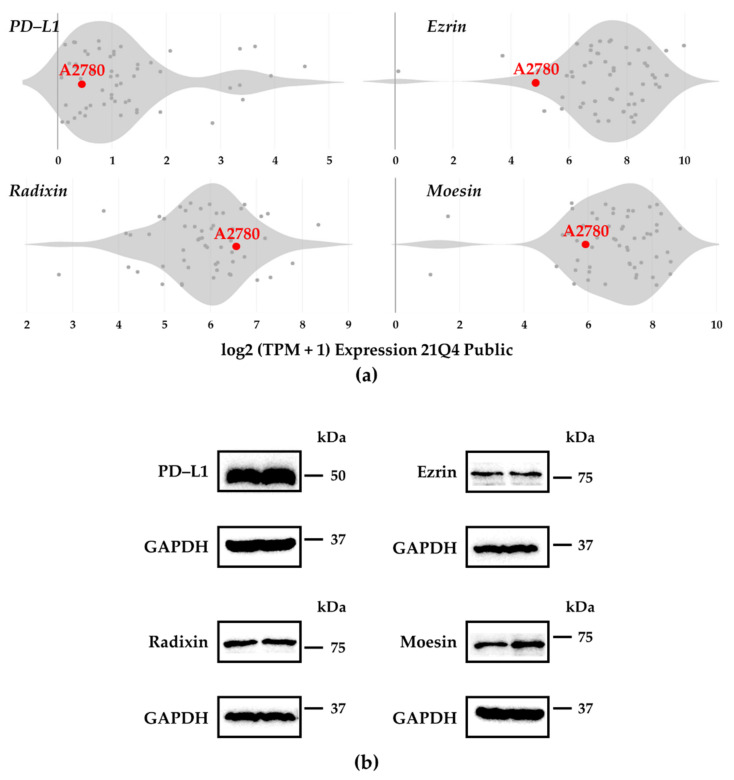
Gene and protein expression patterns of programmed death ligand–1, ezrin, radixin, and moesin in A2780 cells. (**a**) Violin plots showing the median gene expression (log2 (TPM + 1)) of programmed death ligand–1 (PD–L1), ezrin, radixin, and moesin (ERM) in numerous human ovarian adenocarcinoma strains obtained from the database of the Cancer Dependency Map (DepMap) portal data explorer, Broad (2021): DepMap 21Q4 Public. (**b**) Representative western blotting images of PD–L1, ERM, and glyceraldehyde–3–phosphate dehydrogenase (GAPDH) in whole-cell lysates of A2780 cells. Molecular weights are shown in kDa. The data are representative of three independent experiments.

**Figure 2 jcm-11-02457-f002:**
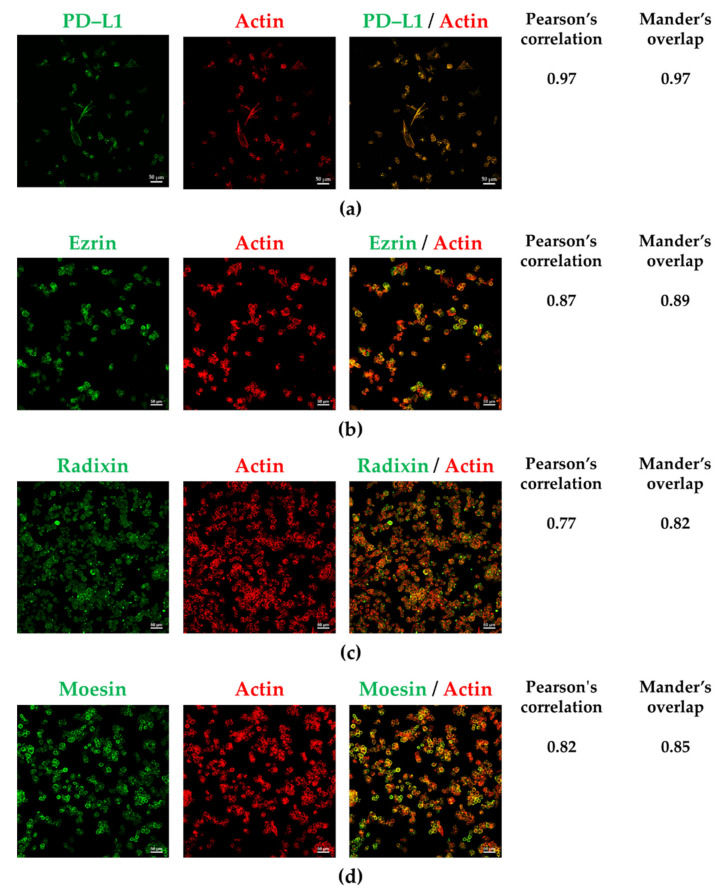
Subcellular localization of programmed death ligand–1, ezrin, radixin, and moesin in A2780 cells. Left panels: Alexa Fluor 488 (green)-labeled (**a**) programmed death ligand–1 (PD–L1), (**b**) ezrin, (**c**) radixin, and (**d**) moesin; middle panels: tetramethylrhodamine (red)-labeled F-actin; right panels: merged fluorescence images. Scale bars: 50 µm. The coefficients of Pearson’s correlation and Mander’s overlap to quantify the colocalization of each protein with actin shown on the right side of each image were expressed as the mean of five independent images.

**Figure 3 jcm-11-02457-f003:**
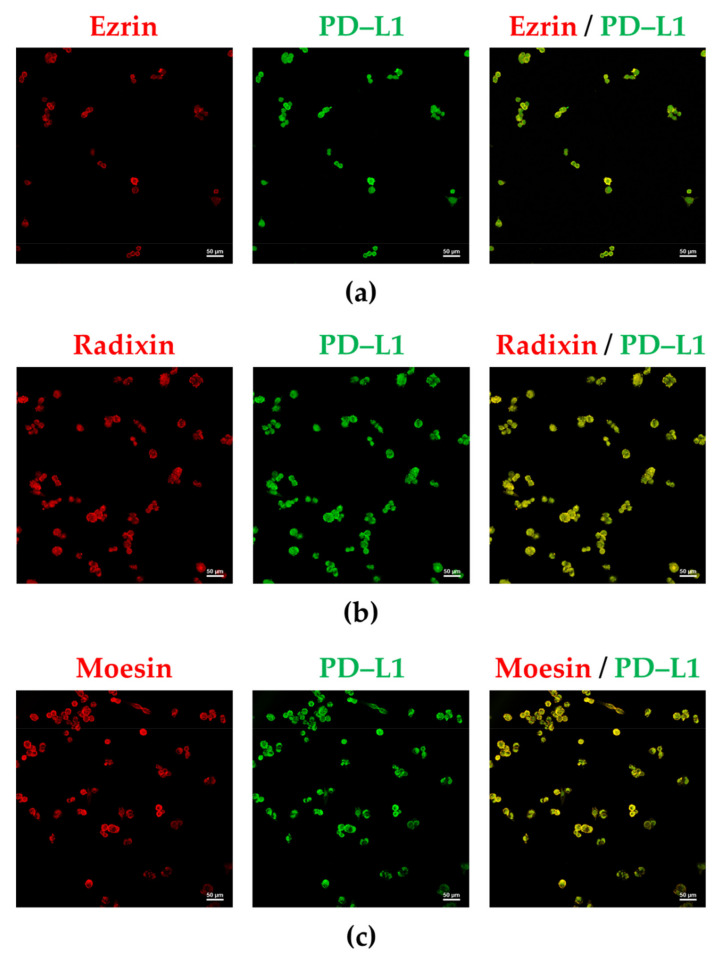
Double immunofluorescence staining for programmed death ligand–1, ezrin, radixin, and moesin in A2780 cells. Left panels: Alexa Fluor 594 (red)-labeled (**a**) ezrin, (**b**) radixin, and (**c**) moesin; middle panels: Alexa Fluor 488 (green)-labeled programmed death ligand–1 (PD–L1); right panels: merged fluorescence images. Scale bars: 50 µm. All images are representative of at least three independent experiments.

**Figure 4 jcm-11-02457-f004:**
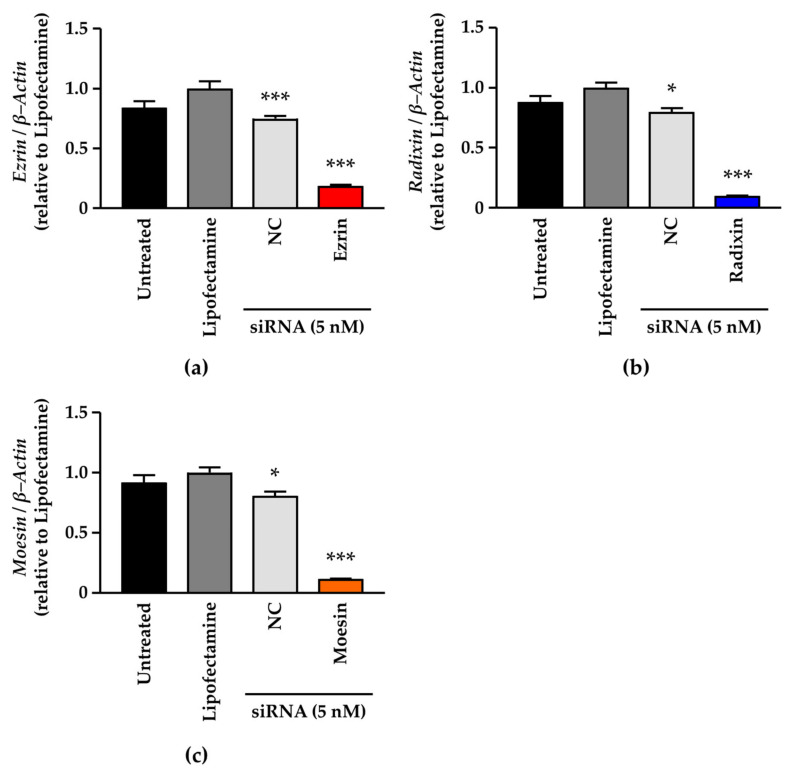
Effect of small interfering RNA against ezrin, radixin, or moesin on each target mRNA expression level in A2780 cells. Cells were cultured for 4 days with the medium (untreated), Lipofectamine, non–targeting control (NC) small interfering (si) RNA, and siRNAs against ezrin, radixin, or moesin at a concentration of 5 nM. Relative mRNA levels of (**a**) ezrin, (**b**) radixin, and (**c**) moesin normalized to β–Actin in cells treated with siRNAs relative to those in cells treated with Lipofectamine alone were measured using real-time reverse transcription–polymerase chain reaction. *n* = 3, *** *p* < 0.001, * *p* < 0.05 vs. Lipofectamine. All data are expressed as the mean ± SEM and were analyzed using one-way ANOVA, followed by Dunnett’s test.

**Figure 5 jcm-11-02457-f005:**
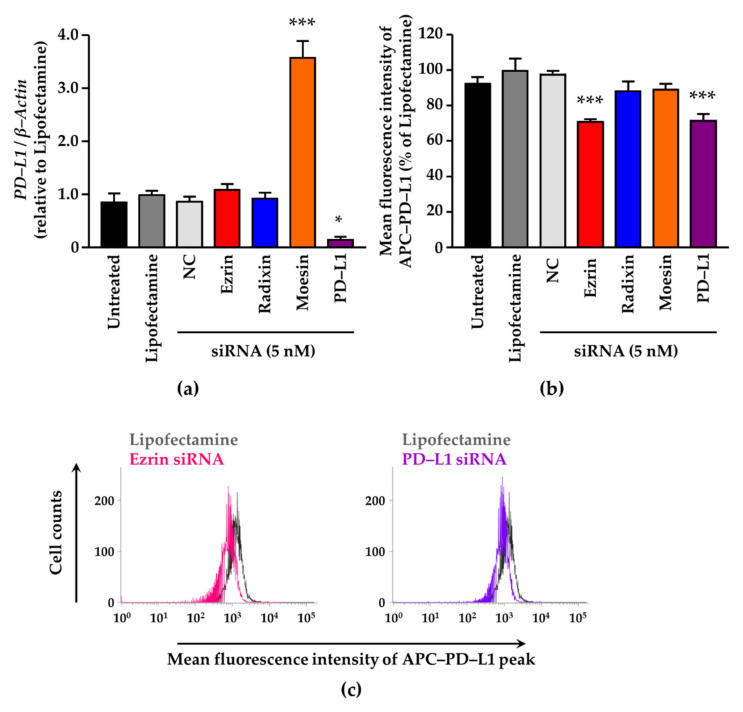
Involvement of ezrin, radixin, and moesin in the gene and cell surface expression of programmed death ligand–1 in A2780 cells. (**a**) Relative mRNA expression level of PD–L1 normalized to β–Actin in the cells treated with small interfering (si) RNAs relative to that in cells treated with the transfection reagent alone was measured using real-time reverse transcription–polymerase chain reaction. *n* = 3, *** *p* < 0.001, * *p* < 0.05 vs. Lipofectamine. (**b**) Relative mean fluorescence intensity of allophycocyanin (APC)-labeled PD–L1 on the plasma membrane of A2780 cells normalized to that of Lipofectamine alone. *n* = 3, *** *p* < 0.001 vs. Lipofectamine. (**c**) Typical histograms of the mean fluorescence intensities of APC-labeled PD–L1 on the plasma membrane of A2780 cells, as measured by flow cytometric analysis. All data are expressed as the mean ± SEM and were analyzed using one-way ANOVA, followed by Dunnett’s test. Non–targeting control; NC.

## Data Availability

The datasets used and analyzed in this study are available from Cancer Cell Line Encyclopedia (https://sites.broadinstitute.org/ccle/datasets, accessed on 25 January 2022), DepMap, Broad (2021): DepMap 21Q4 Public. figshare. Dataset (https://doi.org/10.6084/m9.figshare.19139906.v1, accessed on 25 January 2022). Other data is contained within the article and Appendix A.

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
