# Peer review of "Ezrin Contributes to the Plasma Membrane Expression of PD–L1 in A2780 Cells"

_jcm, 2022, doi:10.3390/jcm11092457_

Round 1

Reviewer 1 Report

Authors describe the role of ezrin gene silencing on the regulation of PD-L1 protein expression in the A2870 ovarian cells without altering mRNA expression.

Major comments:

The section Introduction should be corrected and expanded by the authors' reasoning:

  • P1, l40: 5-year survival rate of OC is approximately 30% (doi: 20892/j.issn.2095-3941.2016.0084, DOI: 10.1016/S0140-6736(18)32552-2)
  • P2, l55: Borella et al, 2020 (doi: 3390/diagnostics10030146) describe 10% response rate, not 40%
  • OC is a very heterogenous group of diseases, this has not been addressed.
  • While 90% of OC cases are epithelial, 70% of these are HGSOC and A2870 cells have been misidentified as HGSOC (Domcke et al 2013, DOI: 10.1038/ncomms3126)

Materials and Methods:

  • It is not clear which passages have been used, if experiments were performed in biological replicas, at different time points,…
  • SiRNA treatment is not described in a way that it can be replicated. What type of transfection was performed? SiRNA information is not included.
  • 3: were RNA integrity numbers measured?
  • 5: which ECL REAGENTS
  • 5: information regarding lots of antibodies is missing, whole WB with proof of validation should be included (including positive and negative controls)

Results:

  • Use of one cell line is not adequate, I suggest adding at least one more to prove the results
  • Ezrin expression is not comparable to other cell lines (Figure 1), further questioning if A2870 as appropriate model
  • I also suggest adding a positive control for silencing (GAPDH?)
  • I also suggest comparing expression to the tissue samples

Perhaps also the title of the manuscript could be more specific, mentioning the A2870 cells instead of epithelial ovarian cancer cells.

Author Response

Response to Reviewer 1’s Comments

We would like to thank #Reviewer 1 for many valuable suggestions on our manuscript. We have carefully read your comments and suggestions and have made the corrections in the revised version of manuscript. Detailed responses to your comments are listed below, and we highlighted all changes with word track changes in the file labeled ‘Revised Manuscript with Track Changes’. We hope this revised manuscript would be satisfactory for publication in Journal of Clinical Medicine.

  1. The section Introduction should be corrected and expanded by the authors' reasoning:

P1, l40: 5–year survival rate of OC is approximately 30% (doi: 20892/j.issn.2095–3941.2016.0084, DOI: 10.1016/S0140–6736(18)32552–2)

Reply Comments.

We would like to appreciate #Reviewer 1’s valuable suggestion. According to #Reviewer 1’s comment, we have changed the description for 5–year survival rate of OC and incorporated the reference #Reviewer 1 kindly suggested.

Introduction (Line 42)

5–year survival rate of only 30% [3]

References (Line 317 318)

  1. Reid, B.M.; Permuth, J.B.; Sellers, T.A. Epidemiology of ovarian cancer: a review. Cancer Biol Med 2017, 14, 9–32, doi:10.20892/j.issn.2095–3941.2016.0084.

P2, l55: Borella et al, 2020 (doi: 3390/diagnostics10030146) describe 10% response rate, not 40%

Reply Comments.

We would like to appreciate #Reviewer 1’s valuable suggestion. According to #Reviewer 1’s comment, we have changed the response rate of immune checkpoint blockade therapy in OC and incorporated the reference #Reviewer 1 kindly suggested.

Introduction (Line 58 – 59)

only 10% of patients respond clinically to anti–PD–1/PD–L1 Abs owing to early or acquired resistance [15].

References (Line 349 351)

  1. Borella, F.; Ghisoni, E.; Giannone, G.; Cosma, S.; Benedetto, C.; Valabrega, G.; Katsaros, D. Immune Checkpoint Inhibitors in Epithelial Ovarian Cancer: An Overview on Efficacy and Future Perspectives. Diagnostics 2020, 10, 146.

OC is a very heterogenous group of diseases, this has not been addressed.

Reply Comments.

We would like to appreciate #Reviewer 1’s valuable suggestion. According to #Reviewer 1’s comment, we have incorporated the sentence describing the heterogeneity of OC.

Introduction (Line 38 – 40)

Ovarian cancer is a highly heterogeneous group of malignancy in terms of morphology, molecular alterations, and clinical behavior [2].

References (Line 315 316)

  1. Kim, J.; Park, E.Y.; Kim, O.; Schilder, J.M.; Coffey, D.M.; Cho, C.H.; Bast, R.C., Jr. Cell Origins of High–Grade Serous Ovarian Cancer. Cancers (Basel) 2018, 10, doi:10.3390/cancers10110433.

While 90% of OC cases are epithelial, 70% of these are HGSOC and A2870 cells have been misidentified as HGSOC (Domcke et al 2013, DOI: 10.1038/ncomms3126)

Reply Comments.

We would like to appreciate #Reviewer 1’s valuable suggestion. While Domcke S. et al. (Nature Commun., 4: 2126 (2013)) described that A2780 cells did not resemble a HGSOC, however, A2780 cells are still widely used as epithelial ovarian cells in the fairly recent past. Therefore, we have selected A2780 cells as a model of epithelial ovarian cancer cells. According to #Reviewer 1’s comment, we have changed the following sentences in the Introduction.

Introduction (Line 72 – 73)

The aim of this study was to examine the role of ERM proteins in the plasma membrane expression of PD–L1 in A2780 cells, derived from human epithelial ovarian cancer.

  1. Materials and Methods:

It is not clear which passages have been used, if experiments were performed in biological replicas, at different time points,…

Reply Comments.

We would like to appreciate #Reviewer 1’s valuable suggestion. According to #Reviewer 1’s comment, we have added the passage number of A2780 cells used in this study.

Materials and Methods (Line 78 81)

A2780 cells were cultured in Roswell Park Memorial Institute (RPMI)–1640 medium (FUJIFILM Wako Pure Chemical, Osaka, Japan) supplemented with heat–inactivated 10% fetal bovine serum (BioWest, Nuailé, France) at passage number from 8 up to 18.

SiRNA treatment is not described in a way that it can be replicated.

What type of transfection was performed?

SiRNA information is not included.

Reply Comments.

We would like to appreciate #Reviewer 1’s valuable suggestion. According to #Reviewer 1’s comment, we have added the transfection methods employed and silencer select siRNA information (assay ID).

Materials and Methods (Line 87 88)

delivered through lipofection using Lipofectamine RNAiMAX

Materials and Methods (Line 85 86)

human ezrin (siRNA ID s14795), radixin (siRNA ID s11899), moesin (siRNA ID s8986), PD–L1(siRNA ID s26549)

3: were RNA integrity numbers measured?

Reply Comments.

We would like to appreciate #Reviewer 1’s valuable suggestion. Although RNA Integrity Number must be measured in the case of RNA–Seq using such as next–generation sequencer, we think the measurement of RNA Integrity Number seems to be not required in the general PCR analysis.

5: which ECL REAGENTS

Reply Comments.

We would like to appreciate #Reviewer 1’s valuable suggestion. According to #Reviewer 1’s comment, we have incorporated ECL reagent name employed in this study.

Materials and Methods (Line 121 123)

enhanced chemiluminescence (ECL) Prime Western Blotting Detection Reagent (Cytiva, Tokyo, Japan)

5: information regarding lots of antibodies is missing, whole WB with proof of validation should be included (including positive and negative controls)

Reply Comments.

We would like to appreciate #Reviewer 1’s valuable suggestion. According to #Reviewer 1’s comment, we have performed additional experiments to validate the specificity of Abs used in western blotting, and obtained blots were incorporated into Figure S1. Additionally, we have also checked the specificity of above Abs by adopting a different experimental technique (Immunofluorescence analysis) and incorporated the results of negative fluorescence staining for A2780 cells in confocal laser scanning microscopy analysis as Figure S3 into Supplementary Materials.

Materials and Methods (Line 123 124)

Whole blots to confirm the reactivity of all Abs used in western blotting was shown in Figure S1.

Supplementary Materials (Line 16 23)

Whole western blots to confirm the reactivity of all Abs used in this study.

Figure S1. Whole western blots to confirm the reactivity of all Abs used in this study. The original western blotting membrane to detect the protein expression of programmed death ligand–1 (PD–L1), ezrin, radixin, and moesin as well as the corresponding glyceraldehyde–3–phosphate dehydrogenase (GAPDH). Left lane; Whole cell lysates of A2780 cells, Middle Lane; Whole cell lysates of HEC–151 cells (positive control), Right Lane; Negative control.

Results (Line 161 164)

None of fluorescence signals were detected in A2780 cells incubated with goat anti–rabbit IgG (heavy + light) secondary antibodies conjugated with an Alexa Fluor 488 or an Alexa Fluor 594 without first antibodies against PD–L1 or each ERM (Figure S3).

Supplementary Materials (Line 31 37)

Negative fluorescence staining for A2780 cells in confocal laser scanning microscopy analysis

Figure S3. Negative fluorescence staining for A2780 cells in confocal laser scanning microscopy analysis. Negative fluorescence staining in Figure 3. Left and middle; Fluorescence images of goat anti–rabbit IgG (heavy + light) secondary antibodies conjugated with an Alexa Fluor 488 or an Alexa Fluor 594, respectively, without primary antibodies against PD–L1 or each ERM. Right; Phase–contrast image. Scale bars: 50 μm. All images were captured by confocal laser scanning microscopy.

  1. Results:

Use of one cell line is not adequate, I suggest adding at least one more to prove the results

Reply Comments.

We would like to appreciate #Reviewer 1’s valuable suggestion. As #Reviewer 1 pointed out, we also think it is very important to ascertain whether the present results are replicated in other human epithelial ovarian cancer cell lines. Therefore, we would like to test if Ezrin also regulates the cell surface expression of PD–L1 in the ovarian cancer cells derived from not only human epithelial but also other histological phenotypes in our future papers.

Ezrin expression is not comparable to other cell lines (Figure 1), further questioning if A2870 as appropriate model

Reply Comments.

We would like to appreciate #Reviewer 1’s valuable suggestion. Although the relative mRNA expression of Ezrin in A2780 cells is lower than that of other human ovarian cancer cells, however, the median gene expression (log2 (TPM +1)) level of Ezrin obtained from DepMap is sufficient to analyze the role of Ezrin in the cell surface localization of PD–L1 in this study.

I also suggest adding a positive control for silencing (GAPDH?)

Reply Comments.

We would like to appreciate #Reviewer 1’s valuable suggestion. Because we have achieved the knockdown activity of all the target genes by respective siRNA under the condition where non–targeting control siRNA has no impacts on the genes of our interests as shown in Figure 4, a positive control for silencing is not required in this manuscript.

I also suggest comparing expression to the tissue samples

Reply Comments.

We would like to appreciate #Reviewer 1’s valuable suggestion. As #Reviewer 1 pointed out, we also think it is very important to ascertain whether the results of our present in vitro experiments are also replicated in the human and/or mice tissue samples. In the future study, we should address this issue using in vivo xenograft mice model. According to #Reviewer 1’s fruitful advice, we have incorporated our future plan in the Discussion section.

Discussion (Line 278 281)

In order to ascertain whether the results of our present in vitro experiments are also replicated in in vivo tissue samples, the in vivo experiments employed xenograft model mice should be performed in our future studies.

Perhaps also the title of the manuscript could be more specific, mentioning the A2870 cells instead of epithelial ovarian cancer cells.

Reply Comments.

We would like to appreciate #Reviewer 1’s valuable suggestion. According to #Reviewer 1’s comment, we have changed the title of this manuscript.

Title (Line 2 3)

Ezrin Regulates the Plasma Membrane Expression of PD–L1 in A2780 Cells

Reviewer 2 Report

The authors demonstrate that PD-L1 and all three ERMs were ex-pressed at the mRNA and protein levels in A2780 cells, and that PD-L1 was highly colocalized with ezrin and moesin, but not with radixin, in the plasma membrane, indicating ezrin may be responsible for the cell surface localization of PD-L1. The results are of interest to rationally desinging effetive platform of ICIs against ovarian cancers. 

  1. Please explain why the authors focus on the three ERM proteins in their study?
  2. For "..binds to PD-1 expressed on the surface of activated cytotoxic T cell membranes, and in-47 duces apoptosis of activated T cells, thereby blocking antitumor T cell immunity" the related literature could be referred, ADVANCED FUNCTIONAL MATERIALS, 2021, 31, 2104630.

Author Response

Response to Reviewer 2’s Comments

We would like to thank #Reviewer 2 for the greatest evaluation on our manuscript. We have carefully read your comments and suggestions and have made the corrections in the revised version of manuscript. Detailed responses to your comments are listed below, and we highlighted all changes with word track changes in the file labeled ‘Revised Manuscript with Track Changes’. We hope this revised manuscript would be satisfactory for publication in Journal of Clinical Medicine.

Comment 1. Please explain why the authors focus on the three ERM proteins in their study?

Reply Comments.

We would like to appreciate #Reviewer 2’s valuable suggestion. According to #Reviewer 2’s comment, we have incorporated the sentence to impute the reason why we focused on the three ERM proteins in the Introduction.

Introduction (Line 62 – 65)

In particular, our laboratory has focused on the family of ezrin/radixin/moesin (ERM) proteins, which has a high degree of sequence similarity with each other that cross–link the actin cytoskeleton to several cancer–related transmembrane proteins, ………….

Comment 2.  For "..binds to PD–1 expressed on the surface of activated cytotoxic T cell membranes, and in–47 duces apoptosis of activated T cells, thereby blocking antitumor T cell immunity" the related literature could be referred, ADVANCED FUNCTIONAL MATERIALS, 2021, 31, 2104630.

Reply Comments.

We would like to appreciate #Reviewer 2’s valuable suggestion. According to #Reviewer 2’s comment, we have incorporated following reference into the Introduction and Reference.

Introduction (Line 49 – 50)

interacts with PD–1 expressed on the surface of activated cytotoxic T cell membranes, and induces apoptosis of activated T cells, thereby blocking antitumor T cell immunity [7–9].

Reference (Line 331 – 333)

  1. Li, Q.; Zhao, Z.; Qin, X.; Zhang, M.; Du, Q.; Li, Z.; Luan, Y. A Checkpoint–Regulatable Immune Niche Created by Injectable Hydrogel for Tumor Therapy. Advanced Functional Materials 2021, 31, 2104630, doi:10.1002/adfm.202104630.

Reviewer 3 Report

In the manuscript entitled “Role of Ezrin in the Cell Surface Expression of PD-L1 in Human Epithelial Ovarian Cancer Cells”, the authors investigate the mechanism involving ezrin in the membrane translocation of PD-L1 in A2780 human ovarian cancer cells. Authors conclude that ezrin, more than radixin or moesin, may be responsible for the cell surface localization of PD-L1. Therefore, it could be a potential therapeutic target in this pathology.
Despite these interesting results, in my opinion, the manuscript is premature to be published in the present form and needs some major revisions that should be addressed prior to publication:

1) The authors confirmed by PCR the silencing of each ERM in A2780 and, specifically, the role of ezrin in PD-L1 membrane localization; however, I suggest to the authors to assess if ezrin silencing exert any anti-mitogenic effects in this cell line (e.g. effects of ezrin on cell survival or proliferation of A2780). If these possible effects are already reported in literature in the same cell model please specify in discussion session.

2) As ezrin KO and PD-L1 inhibition reduced cell migration and invasion of several cell models, authors should exert some experiments (wound healing or migration assay) to assess if this is confirmed in their experimental condition on A2780 cells.

3) Gene silencing of moesin significantly increased PD-L1 mRNA levels. Authors suggested it could be due to increase production or release of inflammatory CKs. Authors should confirmed this hypothesis by mRNA or protein expression analysis of CKs in A2780.

4) Please provide western blotting images at higher definition 

Author Response

Response to Reviewer 3’s Comments

We would like to thank #Reviewer 3 for many valuable suggestions on our manuscript. We have carefully read your comments and suggestions and have made the corrections in the revised version of manuscript. Detailed responses to your comments are listed below, and we highlighted all changes with word track changes in the file labeled ‘Revised Manuscript with Track Changes’. We hope this revised manuscript would be satisfactory for publication in Journal of Clinical Medicine.

Comment 1. The authors confirmed by PCR the silencing of each ERM in A2780 and, specifically, the role of ezrin in PD–L1 membrane localization; however, I suggest to the authors to assess if ezrin silencing exert any anti–mitogenic effects in this cell line (e.g. effects of ezrin on cell survival or proliferation of A2780). If these possible effects are already reported in literature in the same cell model please specify in discussion session.

Reply Comments.

We would like to appreciate #Reviewer 3’s valuable suggestion. According to #Reviewer 3’s comment, we have incorporated the results of cell viability assay to check if siRNAs used in this study had any cytotoxicity for A2780 cells into Supplementary Materials as Figure S4.

Results (Line 185 – 186)

None of siRNAs used in this study had any influences on the viability of A2780 cells (Figure S4).

Figure S4 (Line 39 – 67) in Supplementary Materials

Influence of siRNAs against each target gene on the cell viability of A2780 cells

Figure S4. Influence of siRNAs against each target gene on the cell viability of A2780 cells. Cells were treated with the transfection medium (Untreated), transfection reagent (Lipofectamine), nontargeting control (NC) siRNA, and specific siRNAs for ezrin, radixin, moesin, or programmed death ligand–1 (PD–L1) at the concentrations of 5 nM, and then cultured for 4 days. Cell viability of A2780 cells was assessed with the PrestoBlue cell viability reagent. Staurosporine 1.0 μM is included as a positive control to reduce in vitro cell viability. n = 6, ***p < 0.001 vs. Lipofectamine. All data were expressed as the mean ± SEM and analyzed by one–way ANOVA followed by Dunnett’s test.

Materials and Methods for Figure S4

Cell Viability Assay

A2780 cells at a density of 5.0 × 103 cells were cultured in 96–well cell culture plates (Corning, Glendale, AZ, USA) overnight at 37 °C in a humidified atmosphere with 5% CO2 to allow for attachment. Then, cells were treated with siRNAs as described in the Main Manuscript and 1.0 µM of staurosporine (Merck, Darmstadt, Germany), a suitable positive control to damage in vitro cell viability, for 4 days without exchanging medium. Thereafter, cells were incubated with a PrestoBlue Cell Viability Reagent (Thermo Fisher Scientific, Tokyo, Japan), a fast and sensitive assay for assessing cell viability [1,2], at 37 °C for 10 min under humidified conditions with 5% CO2, protected from direct light. After that, fluorescence signals were detected at wavelengths of 560 nm (excitation) and 590 nm (emission) using a Synergy HTX Multi–Mode Microplate Reader (Bio Tek Instrument, Winooski, VT, USA).

References for Figure S4

  1. Lall, N.; Henley–Smith, C.J.; De Canha, M.N.; Oosthuizen, C.B.; Berrington, D. Viability Reagent, PrestoBlue, in Comparison with Other Available Reagents, Utilized in Cytotoxicity and Antimicrobial Assays. Int. J. Microbiol. 2013, 2013, 420601, doi:10.1155/2013/420601.
  2. Boncler, M.; Rozalski, M.; Krajewska, U.; Podsedek, A.; Watala, C. Comparison of PrestoBlue and MTT assays of cellular viability in the assessment of anti–proliferative effects of plant extracts on human endothelial cells. J. Pharmacol. Toxicol. Methods 2014, 69, 9–16, doi:10.1016/j.vascn.2013.09.003.

Comment 2. As ezrin KO and PD–L1 inhibition reduced cell migration and invasion of several cell models, authors should exert some experiments (wound healing or migration assay) to assess if this is confirmed in their experimental condition on A2780 cells.

Reply Comments.

We would like to appreciate #Reviewer 3’s valuable suggestion. As #Reviewer 3 pointed out, we also think it is very important to assess if Ezrin and PD–L1 knockdown reduce cell migration and invasion of A2780 cells using in vitro functional wound healing or migration assay. However, we highly focused on the role of ERM (especially in Ezrin) in the expression of PD–L1 at genetic and cell surface levels in this paper. Therefore, our future study should be addressed to determine whether gene silencing of Ezrin affects the activity of migration and invasion of A2780 cells in addition to the cytotoxic activity of T–cells by co–culturing with A2780 cells.

Comment 3. Gene silencing of moesin significantly increased PD–L1 mRNA levels. Authors suggested it could be due to increase production or release of inflammatory CKs. Authors should confirm this hypothesis by mRNA or protein expression analysis of CKs in A2780.

Reply Comments.

We would like to appreciate #Reviewer 3’s valuable suggestion. According to #Reviewer 3’s comment, we have performed additional experiments to test if gene silencing of moesin increase the mRNA expression levels of interferon (IFN)–ɤ, the most famous inflammatory cytokine and inducer for PD–L1 mRNA and incorporated the sentences and results into Discussion section and Supplementary Materials.

Discussion (Line 266 – 268)

In fact, siRNA against moesin significantly increased the mRNA expression level of IFN–γ in A2780 (Figure S5), which may in turn upregulated PD–L1 mRNA expression.

Figure S5 (Line 68 – 76) in Supplementary Materials

Changes in the mRNA expression level of interferon (IFN)–ɤ by gene silencing of ezrin, radixin, and moesin in A2780 cells

Figure S5. Changes in the mRNA expression level of interferon (IFN)–ɤ by gene silencing of ezrin, radixin, and moesin in A2780 cells. Cells were incubated with the transfection medium (Untreated), transfection reagent (Lipofectamine), nontargeting control (NC) siRNA, and specific siRNAs for ezrin, radixin, or moesin and then cultured for 4 days. Gene expression level of IFN–ɤ mRNA normalized with β–Actin in cells treated with each siRNA relative to that in cells treated with the transfection reagent alone. n = 3, *p < 0.05 vs. Lipofectamine. All data were expressed as the mean ± SEM and analyzed by one–way ANOVA followed by Dunnett’s test.

Comment 4. Please provide western blotting images at higher definition

Reply Comments.

We would like to appreciate #Reviewer 3’s valuable comments. We think the resolutions of western blotting images (Figure 1b) seems to be enough to understand the results of this Figure. However, considering #Reviewer 3’s kind suggestion, we have added the original source images obtained by western blotting in Figure S2 in order to facilitate a better understanding of this data for readers.

Materials and Methods (Line 124 125)

Original source images for all data obtained by immunoblots are given in Figure S2.

Supplementary Materials (Line 24 29)

Original source images for all data obtained by immunoblots.

Figure S2. Original source images for all data obtained by immunoblots. The original western blotting membrane to detect the protein expression of programmed death ligand–1 (PD–L1), ezrin, radixin, and moesin as well as the corresponding glyceraldehyde–3–phosphate dehydrogenase (GAPDH) used as a loading control shown in Figure 1b.

Round 2

Reviewer 1 Report

The heterogenity of OC has not been sufficiently adressed, especially subtypec of HGSOC should be mentioned and which subtype of HGSOC would benefit from PD-L1 inhibitors discussed.

In methods lot of antibodies are still missing while the question of biological replicas has not been answered.

Author Response

Response to Reviewer 1’s Comments

We would like to thank #Reviewer 1 for many valuable suggestions on our manuscript. We have carefully read your comments and suggestions and have made the corrections in the revised version of manuscript. Detailed responses to your comments are listed below, and we highlighted all changes with word track changes in the file labeled ‘Revised Manuscript with Track Changes’. We hope this revised manuscript would be satisfactory for publication in Journal of Clinical Medicine.

  1. The heterogeneity of OC has not been sufficiently addressed, especially subtype of HGSOC should be mentioned and which subtype of HGSOC would benefit from PD-L1 inhibitors discussed.

Reply Comments.

We would like to appreciate #Reviewer 1’s valuable suggestion. According to #Reviewer 1’s comment, we have further modified the sentence in the Introduction and Discussion regarding the heterogeneity of OC.

Introduction (Line 39 – 45)

Ovarian cancer is morphologically categorized into two major types: non-epithelial and epithelial group, the latter of which accounts for the vast majority of ovarian cancer cases [3]. Based on the clinicopathologic factors, epithelial malignancy is further divided into two biologically distinct groups: high–grade serous ovaria carcinoma (HGSOC), responsible for more than 70% of ovarian cancer death and over 80% of advanced–stage ovarian cancer diagnoses, and non–HGSOC [2].

Introduction (Line 64 – 67)

Although tumor genomic, transcriptomic, and immune profiling were characterized in patients with HGSOC [16,17], however, it has yet to be determined whether certain sub-types of ovarian cancer are more sensitive to ICIs.

References (Line 358 – 364)

  1. Weberpals, J.I.; Pugh, T.J.; Marco-Casanova, P.; Goss, G.D.; Andrews Wright, N.; Rath, P.; Torchia, J.; Fortuna, A.; Jones, G.N.; Roudier, M.P.; et al. Tumor genomic, transcriptomic, and immune profiling characterizes differential response to first-line platinum chemotherapy in high grade serous ovarian cancer. Cancer Medicine 2021, 10, 3045-3058, doi:10.1002/cam4.3831.
  2. Corvigno, S.; Burks, J.K.; Hu, W.; Zhong, Y.; Jennings, N.B.; Fleming, N.D.; Westin, S.N.; Fellman, B.; Liu, J.; Sood, A.K. Immune microenvironment composition in high-grade serous ovarian cancers based on BRCA mutational status. J. Cancer Res. Clin. Oncol. 2021, 147, 3545-3555, doi:10.1007/s00432-021-03778-1.

  1. In methods lot of antibodies are still missing while the question of biological replicas has not been answered.

Reply Comments.

We would like to appreciate #Reviewer 1’s valuable suggestion. According to #Reviewer 1’s comment, we have incorporated the Lot No. of all the antibodies used in this study. We are terribly sorry for missing your comments regarding the Lot No. of antibodies at the time of Revise#1.

Supplementary Materials (Line 81 82)

Table 2. List of antibodies.

Antibodies

Manufacturer

Cat. No.

Dilution

Lot No.

rabbit anti–ezrin

Cell Signaling Technology

3145

1:2,000 (WB)

1:50 (IF)

3

rabbit anti–radixin

Gene Tex

GTX105408

1:2,000 (WB)

1:100 (IF)

39939

rabbit anti–moesin

Cell Signaling Technology

3150

1:2,000 (WB)

1:50 (IF)

2

Alexa Fluor 488–conjugated rabbit anti–PD–L1

Cell Signaling Technology

25048

1:50 (IF)

2

Alexa Fluor 488–conjugated goat anti–rabbit IgG

Thermo Fisher Scientific

R37116

1:25 (IF)

2277676

Alexa Fluor 594–conjugated goat anti–rabbit IgG

Thermo Fisher Scientific

R37117

1:25 (IF)

2237275

HRP–conjugated rabbit anti–PD–L1

Cell Signaling Technology

51296s

1:1,000 (WB)

5

mouse anti–GAPDH

Merck

MAB374

1:20,000 (WB)

3432602

HRP–conjugated anti–rabbit IgG (heavy + light)

SeraCare Life Sciences

5220–0336

1:10,000 (WB)

10468833

HRP–conjugated anti–mouse IgG (heavy + light)

SeraCare Life Sciences

5220–0341

1:10,000 (WB)

10387619

APC–conjugated mouse anti–human PD–L1

BioLegend

329708

4.0 μg/test (FC)

B311349

Reviewer 3 Report

The authors have responded satisfactorily to my previous requests.

Author Response

Response to Reviewer 3’s Comments

We would like to thank #Reviewer 3 for the valuable suggestion and greatest evaluation on our manuscript. Again, thank you for your time.

This manuscript is a resubmission of an earlier submission. The following is a list of the peer review reports and author responses from that submission.